# CD28 expression is required after T cell priming for helper T cell responses and protective immunity to infection

Michelle A Linterman[1,2]*[†], Alice E Denton[2,3], Devina P Divekar[1,2][‡], Ilona Zvetkova[4], Leanne Kane[5], Cristina Ferreira[6], Marc Veldhoen[6], Simon Clare[5], Gordon Dougan[5], Marion Espéli[1,2][§], Kenneth GC Smith[1,2]

[1]Cambridge Institute for Medical Research, University of Cambridge School of Clinical Medicine, Cambridge, United Kingdom; [2]Department of Medicine, University of Cambridge School of Clinical Medicine, Cambridge, United Kingdom; [3]Cancer Research UK Cambridge Institute, University of Cambridge, Cambridge, United Kingdom; [4]University of Cambridge Metabolic Research Laboratories, Institute of Metabolic Science, Addenbrooke's Hospital, Cambridge, United Kingdom; [5]Wellcome Trust Genome Campus, Wellcome Trust Sanger Institute, Cambridge, United Kingdom; [6]Babraham Research Campus, Babraham Institute, Cambridge, United Kingdom

*For correspondence: Michelle. Linterman@babraham.ac.uk

Present address: [†]Lymphocyte Signalling and Development, Babraham Institute, Cambridge, United Kingdom; [‡]Biomedical Research Centre, School of Biological Sciences, University of East Anglia, Norwich, United Kingdom; [§]UMR-S996, LabEx LERMIT, Clamart, France

Competing interests: The authors declare that no competing interests exist.

**Abstract** The co-stimulatory molecule CD28 is essential for activation of helper T cells. Despite this critical role, it is not known whether CD28 has functions in maintaining T cell responses following activation. To determine the role for CD28 after T cell priming, we generated a strain of mice where CD28 is removed from CD4$^+$ T cells after priming. We show that continued CD28 expression is important for effector CD4$^+$ T cells following infection; maintained CD28 is required for the expansion of T helper type 1 cells, and for the differentiation and maintenance of T follicular helper cells during viral infection. Persistent CD28 is also required for clearance of the bacterium *Citrobacter rodentium* from the gastrointestinal tract. Together, this study demonstrates that CD28 persistence is required for helper T cell polarization in response to infection, describing a novel function for CD28 that is distinct from its role in T cell priming.

## Introduction

T cell activation requires two signals: peptide in the context of the major histocompatibility complex (MHC) interacting with the T cell receptor (TCR), and a co-stimulatory signal (*Lafferty and Cunningham, 1975*). The binding of CD28 to its ligands, CD80/CD86, initiates a transcriptional program that enables effective T cell proliferation and differentiation. In the absence of CD28 signaling T cells fail to respond, and become anergic (*Diehn et al., 2002*; *Riley et al., 2002*). CD28 signaling is essential for multiple facets of CD4$^+$ T cell activation (*Harding et al., 1992*), including proliferation, survival (*Boise et al., 1995*), glucose metabolism (*Frauwirth et al., 2002*), and migration (*Marelli-Berg et al., 2007*). Consequently, CD28-deficient mice have reduced expansion of effector CD4$^+$ T cells and do not form T helper type 1 (Th1) and T follicular helper (Tfh) cells after infection or immunization. Tfh cells provide growth and differentiation signals to germinal center B cells, enabling them to exit the germinal center as long-lived plasma cells or memory B cells (*Crotty, 2011*). Th1 cells are formed during a Type 1 immune response and facilitate the clearance of intracellular pathogens (*Zhu and Paul, 2010*). Impaired formation of Tfh and Th1 cells results in impaired cellular and humoral immunity to foreign antigens in CD28-deficient mice (*Shahinian et al., 1993*; *Green et al., 1994*; *Ferguson et al., 1996*; *King et al., 1996*;

**eLife digest** Invasion by a bacterium or virus typically activates a mammalian host's immune system to eliminate the pathogen. The cells of the so-called 'innate immune system' are the body's first line of defense against infection, and these cells patrol the organs and tissues in an effort to locate and eliminate pathogens quickly. The innate immune response is rapid and non-specific, but often cannot completely clear an infection. When necessary, innate immune cells will escalate the immune response by activating the second branch of the immune system, called the 'adaptive immune system'. This specifically targets and eradicates an invading pathogen.

T cells are essential components of the adaptive immune system, and these cells can be readily distinguished from other types of cell by proteins called T cell receptors (or TCRs) found on their surface. There are also different types of T cell, each with a specific function. T helper cells, for example, help other adaptive immune cells to mature and activate, which involves these immune cells proliferating and developing into more specialized cells.

For a T cell to activate, two events must occur at the same time. First, the TCR must recognize and bind to a fragment of the pathogen that is presented to it by an innate immune cell. And second, 'co-stimulatory molecules' present on the surfaces of both the T cell and the same innate immune cell must interact. Using these two signals to activate a T cell helps to ensure the adaptive immune response is not 'unleashed' unnecessarily.

Co-stimulatory molecules have become popular targets for therapies aimed at treating autoimmune disorders—where the immune system attacks and destroys the body's own tissues. One of the most well studied co-stimulatory molecules expressed by T cells is called CD28; however, it remained unknown whether CD28 is involved in any processes after T cell activation.

Now, Linterman et al. reveal that the CD28 co-stimulatory molecule plays a number of roles in addition to T cell activation. For example, a newly developed mouse model showed that CD28 must remain on the surface of T helper cells after they have been activated for these cells to effectively specialize. Linterman et al. also discovered that CD28 helps different T helper cell subtypes to develop.

Linterman et al. demonstrate that CD28 is critical throughout a host's response to infection, and suggest that if CD28 is lost on activated T cells (which happens during aging, HIV infection and autoimmune diseases) the responses of T helper cells become limited. Furthermore, these findings reveal that treatments that target the CD28 co-stimulatory molecule will also affect on-going immune responses.

---

*Mittrucker et al., 1996*; *Walunas et al., 1996*; *Walker et al., 1999*; *Bertram et al., 2002*). From this extensive body of literature, it is clear that CD28 signaling plays an essential role in CD4[+] T cell priming and the formation of effector T cell subsets, but this has made it difficult to determine whether CD28 signaling also contributes to the ongoing immune response, as downstream phenotypes may be due to defective initial activation of CD28-deficient T cells. The expression of CD28 and its ligands increases progressively after priming, raising the possibility that CD28 signaling is utilized during the effector phase of the immune response.

Despite the central role of CD28 signaling in T cell biology, a role for sustained CD28 expression in CD4[+] T cell subset differentiation and maintenance has not been described. This is likely to be because the role for CD28 after T cell priming is difficult to determine experimentally. CD28-deficient helper T cells do not become activated, and thus cannot be used to evaluate the role for CD28 after T cell priming. Moreover, CD28 has two ligands, CD80 and CD86, which it shares with the inhibitory receptor cytotoxic T-lymphocyte antigen 4 (CTLA-4). Consequently, manipulating CD80/86 cannot be used to draw conclusions on the role of CD28 alone, as CTLA-4 ligation will also be affected. The role for CD28 on activated T cells is particularly pertinent as CD28 expression is lost on a proportion of activated T cells with age, in HIV infection, and in the number of immune disorders in primates (but not rodents) (*Weng et al., 2009*; *Aberg, 2012*; *Broux et al., 2012*), and the functional consequence of this loss of CD28 is unknown. In addition, the CD28 pathway is an area of considerable research interest as it can be manipulated by a number of therapeutics: blocking this pathway is being used to treat autoimmune and inflammatory disease and to prevent transplant rejection (*Salomon and Bluestone, 2001*). In

contrast, engaging this pathway can expand anti-tumor T cells and regulatory T cells (*Carreno et al., 2005*). Because of the considerable therapeutic potential of this pathway, understanding its biology is of direct clinical relevance.

Studies have suggested that CD28 may play a role following T cell priming, for example blocking CD28 ligands during an ongoing immune response can impair the germinal center response (*Han et al., 1995*), prolong graft survival, and suppress autoimmunity (*Salomon and Bluestone, 2001*). These studies support the possibility for a role for CD28 in an ongoing immune response. However, attributing these phenotypes to a role for continued CD28 expression in established immune responses is not possible because CD28 signaling is blocked on all cells, and not only on previously activated T cells. To determine whether there is a role for CD28 signaling after T cell activation, we generated a strain of *Cd28^{flox/flox} Ox40^{cre/+}* mice where CD28 expression is lost after T cell priming. We show that the numbers of both Tfh and Th1 cells are reduced in *Cd28^{flox/flox} Ox40^{cre/+}* mice after influenza A virus infection, although, surprisingly, the requirement for CD28 on each cell type is distinct. Tfh differentiation requires CD28 ligation during interactions of primed T cells with B cells and fully differentiated Tfh cells require CD28 expression for their survival. By contrast, Th1 cells do not require CD28 for their maintenance, but do for their expansion following T cell activation. Furthermore, *Cd28^{flox/flox} Ox40^{cre/+}* mice are unable to clear *Citrobacter rodentium* from their gastrointestinal tract following oral infection. This demonstrates that CD28 expression is required after T cell priming for intact effector CD4+ T cell responses during infection.

## Results

### CD28^{flox/flox} Ox40^{cre/+} mice have intact early T cell activation

To generate a strain of mice where CD28 is lost after T cell priming, we took advantage of the expression pattern of OX40 (encoded by the *Tnfrsf4* gene), a co-stimulatory molecule that is induced after T cell priming (*Mallett et al., 1990*; *Gramaglia et al., 1998*). A strain of mice that expresses cre-recombinase from the *Ox40* locus (*Klinger et al., 2009*) was crossed with *Cd28^{flox/flox}* mice. In these mice, we expect that cre-recombinase will be expressed after T cell priming, and CD28 signaling will be intact for initial T cell priming, then removed. To test this, we bred *Cd28^{flox/flox} Ox40^{cre/+}* mice with OT-II transgenic mice, which express a T cell receptor specific for peptide 323–339 of chicken ovalbumin (OVA). We assessed whether CD28 was lost after T cell activation and if early CD28-dependent events, proliferation and production of the mitogenic cytokine interleukin-2 (IL-2) (*Harding et al., 1992*), occur in OT-II *Cd28^{flox/flox} Ox40^{cre/+}* cells. OT-II *Cd28^{+/flox} Ox40^{cre/+}* control or OT-II *Cd28^{flox/flox} Ox40^{cre/+}* T cells labeled with cell trace violet were transferred into CD45.1 C57BL/6 mice, and immunized with OVA. In the absence of immunization, all cells expressed CD28, and did not divide (*Figure 1A*). 48 hr following immunization both OT-II *Cd28^{+/flox} Ox40^{cre/+}* control and OT-II *Cd28^{flox/flox} Ox40^{cre/+}* T cells had undergone up to four cell divisions, as measured by dilution of cell trace violet, and around 30% of activated OT-II *Cd28^{flox/flox} Ox40^{cre/+}* cells had lost CD28 expression (*Figure 1A*). Both OT-II *Cd28^{+/flox} Ox40^{cre/+}* control and OT-II *Cd28^{flox/flox} Ox40^{cre/+}* T cells produced IL-2, consistent with activation via CD28 (*Figure 1B,D*). Importantly, IL-2 was produced by *Cd28^{flox/flox} Ox40^{cre/+}* T cells irrespective of whether they have maintained (CD28+) or lost CD28 expression (CD28−), suggesting that CD28− cells have indeed been activated through CD28 signaling prior to induction of OX40cre (*Figure 1C,D*). There was also equivalent induction of the Inducible T-cell COStimulator (ICOS), a molecule whose expression is dependent on CD28 signaling (*McAdam et al., 2000*), and the T cell activation marker CD44 on OT-II *Cd28^{+/flox} Ox40^{cre/+}* and OT-II *Cd28^{flox/flox} Ox40^{cre/+}* T cells (*Figure 1E,F*). Furthermore, both ICOS and CD44 were expressed at similar levels on CD28+ and CD28− cells from the OT-II *Cd28^{flox/flox} Ox40^{cre/+}* T cell population (*Figure 1E,F*). These data demonstrate that *Cd28^{flox/flox} Ox40^{cre/+}* T cells can be primed and subsequently divide, produce IL-2, and upregulate activation markers.

We then assessed the T cell phenotype of non-TCR transgenic *Cd28^{flox/flox} Ox40^{cre/+}* mice. OX40cre expression is largely restricted to the CD4+ T cell compartment (*Klinger et al., 2009*) and around half of the activated (CD44^{high}) cells expressed Cre and had lost CD28 expression (*Figure 1G*). Furthermore, 15–20% of naïve (CD44^{low}) cells have expressed OX40cre and had lost CD28 (*Figure 1G*). It has previously been demonstrated that 'naïve' cells that had switched on OX40cre have received stronger TCR signals in the thymus, and have a partially activated phenotype that is distinct from OX40cre-negative CD44^{low} cells (*Klinger et al., 2009*). Consistent with this report, in splenocytes from OT-II *Cd28^{flox/flox} Ox40^{cre/+}* mice, where ~70% of CD4+ T cells recognize peptide 323–339 of OVA (a foreign antigen that

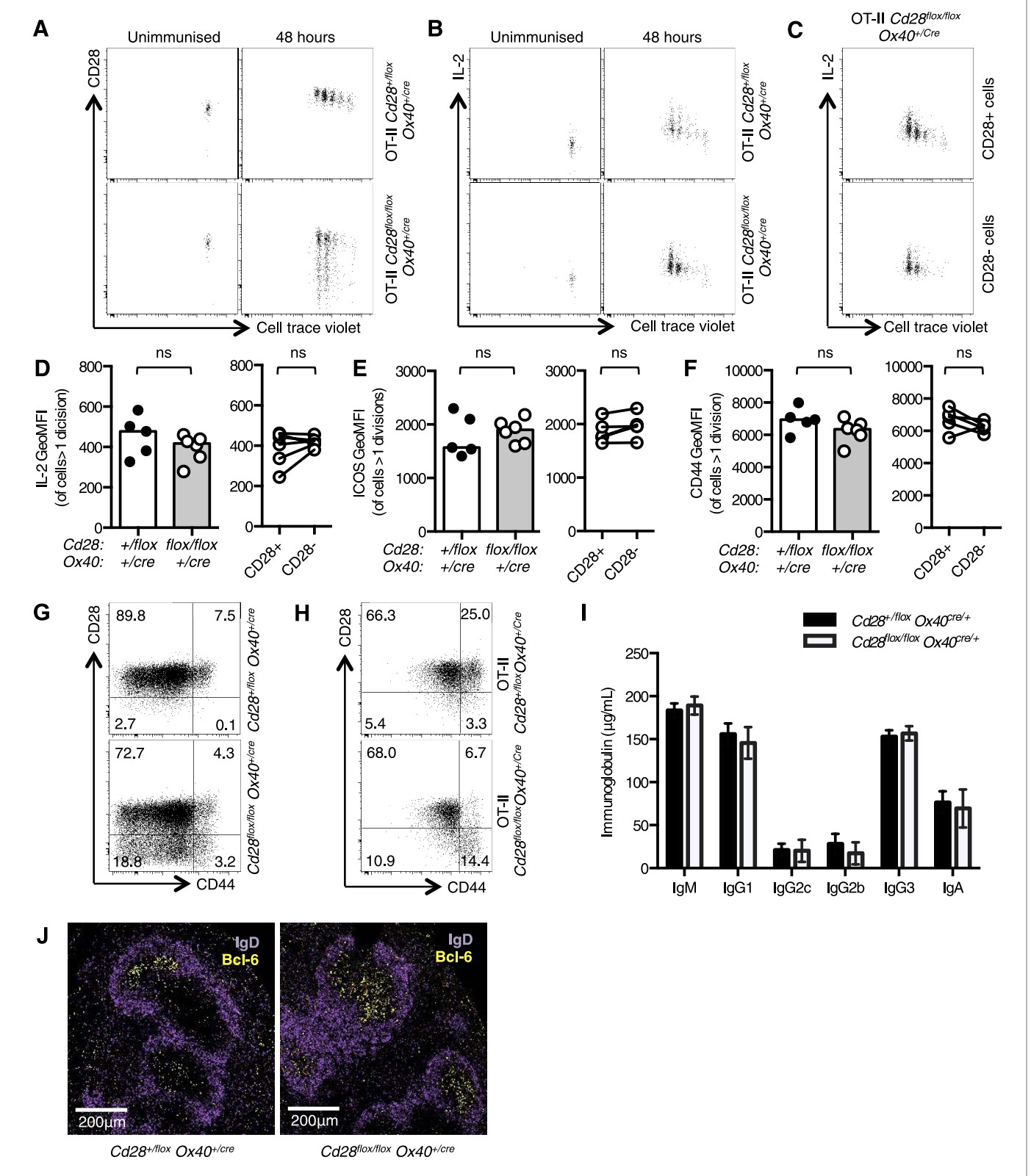

**Figure 1**. *Cd28^{flox/flox} Ox40^{cre/+}* mice lose CD28 expression after T cell priming. 1 × 10^5 OT-II T cells were labeled with cell trace violet (CTV) and transferred into CD45.1 C57BL/6 hosts and immunized with OVA. The dilution of CTV, CD28 expression (**A**) and IL-2 production (**B**) was assessed in OT-II *Cd28^{flox/flox} Ox40^{cre/+}* T cells and controls, 48 hr following immunization. The production of IL-2 was also assessed in CD28+ (top panel) and CD28- (lower panel)
*Figure 1. Continued on next page*

*Figure 1. Continued*

OT-II cells from OT-II *Cd28^flox/flox Ox40^cre/+*mice (**C**). The expression of IL-2 (**D**) ICOS (**E**) and CD44 (**F**) was quantified for the same experimental system. Flow cytometric dot plots of CD28 and CD44 expression on CD4$^+$ splenocytes from *Cd28^flox/flox Ox40^cre/+* mice (**G**), and OT-II *Cd28^flox/flox Ox40^cre/+* mice and controls (**H**). Basal serum immunoglobulins (**I**) from unmanipulated *Cd28^flox/flox Ox40^cre/+* mice and heterozygous controls. (**J**) Representative confocal immunofluorescence of IgD (purple) and Bcl-6 (yellow) staining in spleen sections taken 7 days after sheep red blood cell immunization of *Cd28^+/flox Ox40^cre/+* and *Cd28^flox/flox Ox40^cre/+* mice. In **D–F** heights of the bars represent the median values. In **I, J**, heights of the bars represent the mean values and error bars represent SEM. For experiments **A–F**, data are representative of six independent experiments with 5–6 mice per group. Experiments (**H** and **I**) are representative of three independent experiments with five mice per group. Experiment (**J**) is representative of three independent experiments with four mice per group. Ns = not significant, *p < 0.05, **p < 0.005, ***p < 0.005.

is not expressed in the thymus), CD28 expression is maintained on CD44$^{low}$ cells (*Figure 1H*). CD28-deficient mice have impaired basal serum titers of IgG1 and IgG2a and germinal center formation after immunization (*Shahinian et al., 1993*; *Ferguson et al., 1996*), by contrast *Cd28^flox/flox Ox40^cre/+* mice had comparable basal serum immunoglobulin (*Figure 1I*) to heterozygous control animals, and formed germinal centers 7 days after sheep red blood cell immunization (*Figure 1J*). These data demonstrate that *Cd28^flox/flox Ox40^cre/+* have intact T cell priming, distinguishing them from CD28-deficient mice, and are a novel tool to assess the role for CD28 signaling after T cell activation.

## *Cd28^flox/flox Ox40^cre/+* mice had fewer Tfh and Th1 cells after influenza infection

To assess the implications of loss of CD28 after T cell activation, we infected *Cd28^flox/flox Ox40^cre/+* and heterozygous control mice intranasally (I.N.) with influenza A virus (HKx31), which causes an acute localized infection in the respiratory tract that generates robust anti-viral T and B cell responses in the draining mediastinal lymph node (medLN) and the lungs. In the early phase of the infection, 5 days post infection, ~50% of activated CD4$^+$ T cells had lost CD28 expression (*Figure 2A*), IL-2 production was intact (*Figure 2B*), and there were comparable proportions of proliferating cells in the medLN (*Figure 2C*) of *Cd28^flox/flox Ox40^cre/+* mice compared to controls, and between CD28$^+$ and CD28$^-$ cells. At this time point, there were comparable numbers of CXCR5$^+$Bcl-6$^+$CD4$^+$ Tfh-precursors (*Figure 2D*) and IFNγ$^+$CD44$^{high}$CD4$^+$ Th1 cells (*Figure 2E*) in the draining lymph node of *Cd28^flox/flox Ox40^cre/+* mice and heterozygous controls, demonstrating that T cell priming was intact. However, 12 days post infection, *Cd28^flox/flox Ox40^cre/+* mice had reduced numbers of CXCR5$^+$Bcl-6$^+$CD4$^+$ Tfh cells (*Figure 2D*) and IFNγ$^+$CD44$^{high}$CD4$^+$ Th1 cells (*Figure 2E*) in the medLN and Th1 cells in the lungs (*Figure 2F*) compared to control mice. Corresponding to a deficit in Tfh cells, there were fewer Bcl-6$^+$Ki-67$^+$B220$^+$ germinal center B cells (*Figure 2G*) in *Cd28^flox/flox Ox40^cre/+* mice. Expression of CD28 on Tfh in the medLN and Th1 cells in the lung was determined 12 days following influenza infection: 30–40% of Tfh cells (*Figure 2H*) and 55–65% of lung Th1 cells (*Figure 2I*) had lost CD28 expression in *Cd28^flox/flox Ox40^cre/+* mice.

To determine if this phenotype was cell intrinsic, we generated mixed chimeras with a 1:1 ratio of CD45.1 *Cd28^+/flox Ox40^cre/+*: CD45.2 *Cd28^flox/flox Ox40^cre/+* bone marrow, and control chimeras with CD45.1 *Cd28^+/flox Ox40^cre/+*: CD45.2 *Cd28^+/flox Ox40^cre/+* bone marrow. 8 weeks after reconstitution, these chimeras were infected I.N. with influenza A virus. In control chimeras, 12 days post infection, CD45.1 and CD45.2 cells contributed equally to the Tfh cell pool and Th1 cell population in the draining lymph node and lung as expected (*Figure 2—figure supplement 1*). In contrast, in the CD45.1 *Cd28^+/flox Ox40^cre/+*: CD45.2 *Cd28^flox/flox Ox40^cre/+* chimeras the CD45.2$^+$ cells barely contributed to the Tfh or Th1 cell populations (*Figure 2—figure supplement 1*). This demonstrated that T cells derived from *Cd28^flox/flox Ox40^cre/+* stem cells are outcompeted during the response to influenza A virus. Taken together, these data suggest that continued, cell-intrinsic CD28 signaling in CD4$^+$ T cells is required for the development of a full population of Tfh and Th1 cells in response to influenza A virus infection.

## Continued CD28 expression is required to establish the Tfh population

The observation that fewer Tfh cells form after influenza infection in *Cd28^flox/flox Ox40^cre/+* mice suggests that their formation and/or maintenance is impaired in the absence of CD28. Tfh differentiation is a stepwise process that requires multiple receptor–ligand interactions (*Linterman et al., 2012*). T cells must first be primed by dendritic cells, a process that requires both peptide-MHC:TCR interactions

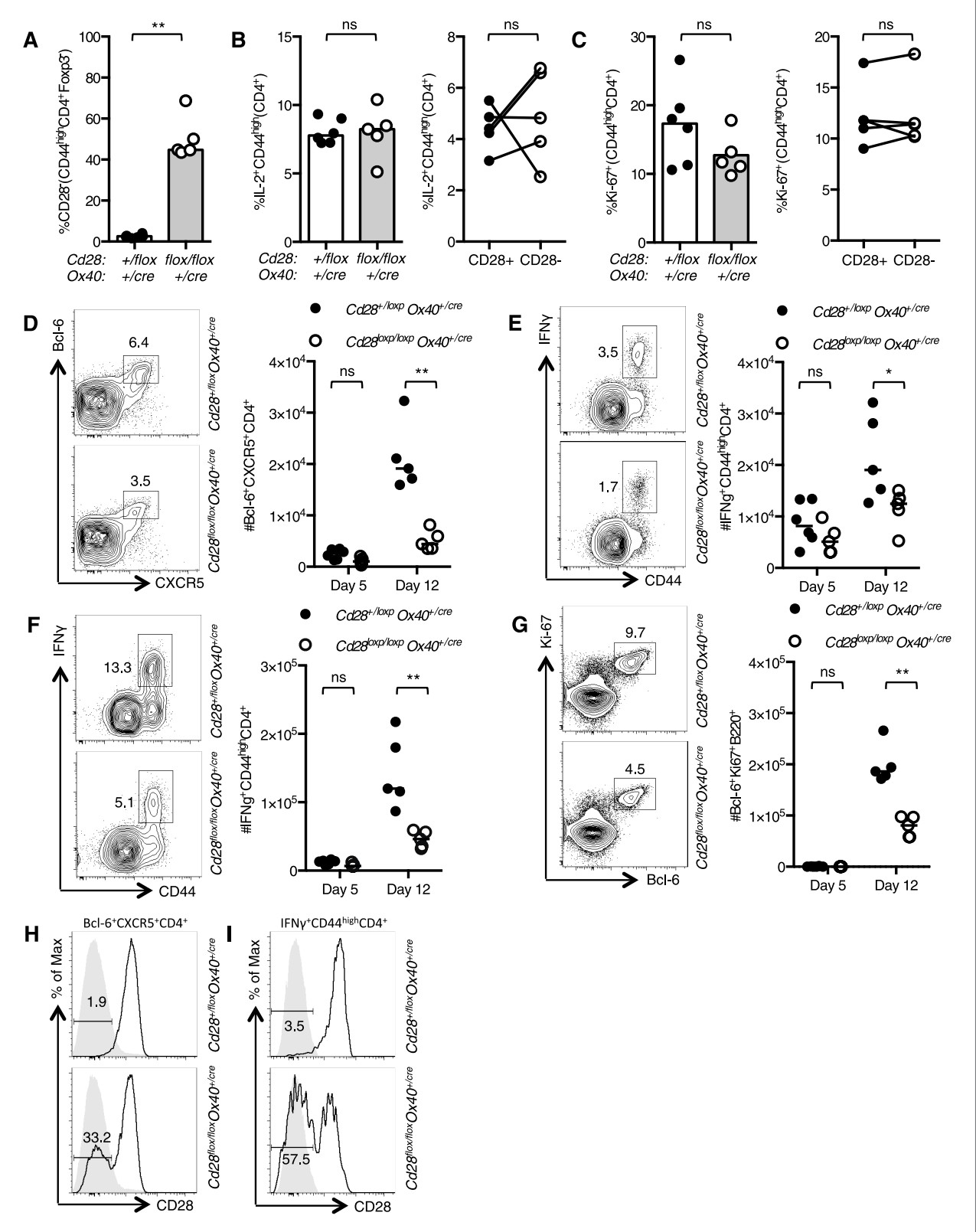

**Figure 2**. The cellular immune response to Influenza A infection is impaired in *Cd28^flox/flox^ Ox40^cre/+^ mice*. *Cd28^flox/flox^ Ox40^cre/+^* mice and heterozygous controls were infected I.N. with 10⁴ plaque-forming units of influenza A virus. Graphs showing proportion of activated CD4 T cells that have lost CD28 (**A**) are producing IL-2 (**B**) and are Ki-67⁺ (**C**) in the medLN 5 days post infection. Representative flow cytometric contour plots from day 12 post

*Figure 2. Continued on next page*

*Figure 2. Continued*

infection and graphs show the number of medLN Bcl-6$^+$CXCR5$^+$CD4$^+$Foxp3$^-$ Tfh cells (**D**), IFNγ$^+$CD44$^{high}$CD4$^+$ Th1 cells in the medLN (**E**) and lung (**F**) and medLN Bcl-6$^+$Ki67$^+$B220$^+$ germinal center B cells (**G**) at day 5 and day 12 following infection in Cd28$^{+/flox}$ Ox40$^{cre/+}$ and Cd28$^{flox/flox}$ Ox40$^{cre/+}$ mice. Histograms of CD28 expression on medLN Bcl-6$^+$CXCR5$^+$CD4$^+$Foxp3$^-$ Tfh cells (**H**) and lung IFNγ$^+$CD44$^{high}$CD4$^+$ Th1 cells (**I**) 12 days after influenza infection. Heights of the lines on graphs represent the median values. Data are representative of three independent experiments with 5–8 mice per group. Ns = not significant, *p < 0.05, **p < 0.005, ***p < 0.005.
The following figure supplement is available for figure 2:

**Figure supplement 1**. *Tfh* and *Th1* cells from *Cd28$^{flox/flox}$ Ox40$^{cre/+}$* origin are outcompeted in mixed bone marrow chimeras.

and CD28 ligation, which facilitates Bcl-6 expression. For full Tfh differentiation, primed T cells then need a second round of antigen presentation from B cells to stabilize Bcl-6 expression and generate Tfh cells that are capable of migrating into B cell follicles and supporting the germinal center response (*Deenick et al., 2010*; *Baumjohann et al., 2011*; *Choi et al., 2011*).

To confirm that CD28 is required for Tfh development after initial T cell priming, we used an adoptive transfer system that has been previously used to characterize Tfh differentiation (*Baumjohann et al., 2011*). OT-II *Cd28$^{+/flox}$ Ox40$^{cre/+}$* control or OT-II *Cd28$^{flox/flox}$ Ox40$^{cre/+}$* T cells were transferred into CD45.1 C57BL/6 mice and immunized with OVA. 3.5 days after immunization ~50% of the transferred OT-II *Cd28$^{flox/flox}$ Ox40$^{cre/+}$* T cells had lost CD28 expression (*Figure 3A*) and had undergone numerous cell divisions (*Figure 3B*). OT-II *Cd28$^{flox/flox}$ Ox40$^{cre/+}$* T cells formed fewer CXCR5$^+$Bcl-6$^+$ Tfh-precursors than heterozygous control OT-II T cells (*Figure 3C,E*). Furthermore, 90% of the CD4$^+$ T cells that had a CXCR5$^+$Bcl-6$^+$ pre-Tfh phenotype were CD28$^+$ (*Figure 3D,E*), demonstrating that T cells that have lost CD28 are less able to differentiate into Tfh cells. Both apoptosis and cell division were similar between CD28$^+$ and CD28$^-$ cells (*Figure 3F,G*). However, ICOS expression was reduced on pre-Tfh cells that had lost CD28 (*Figure 3H*). These data demonstrate that CD28 expression is required after T cell activation to establish the Tfh population.

An independent experimental system was used to confirm that Tfh differentiation requires CD28 signaling after T cell priming: control CD45.1 OT-II T cells were transferred into CD45.2 hosts and immunized with OVA. 48 hr after immunization antibodies that block the CD28 ligands, CD80 and CD86 were administered, and pre-Tfh formation was assessed 1.5 days later (3.5 days after initial immunization). This impaired the formation of CXCR5$^+$Bcl-6$^+$CD4$^+$ Tfh-precursors (*Figure 3I*), supporting data from *Cd28$^{flox/flox}$ Ox40$^{cre/+}$* mice showing that CD28 signaling is required after T cell priming for Tfh differentiation, at a time point where B cells are the major antigen presenting cells for Tfh differentiation.

## CD28 is required for the maintenance of Tfh cells

The germinal center is rich in CD28 ligands, suggesting that there may be a conserved CD28:CD28L interaction within this microenvironment. We observed that the percentage of Tfh cells that had lost CD28 expression decreased over time during the response to influenza (*Figure 4A,B*), suggesting that Tfh maintenance is impaired by the loss of CD28. Consistent with this, there was an increase in the number of apoptotic Tfh cells 14 days after influenza infection in *Cd28$^{flox/flox}$ Ox40$^{cre/+}$* mice compared with heterozygous controls (*Figure 4C*), and this was most pronounced in the cells that had lost CD28 expression (*Figure 4C*). There was also an increase in proliferation of Tfh cells in *Cd28$^{flox/flox}$ Ox40$^{cre/+}$* mice (*Figure 4D*), although this was equivalent in CD28$^+$ and CD28$^-$ Tfh cells, suggesting a broader, perhaps compensatory increase in proliferation, rather than the one that can be specifically attributed to the loss of CD28. There was also a decrease in expression of two molecules downstream of CD28; the pro-survival molecule Bcl-$_{XL}$ (*Figure 4E*) and the costimulatory molecule ICOS (*Figure 4F*) in Tfh cells that had lost CD28. Expression of the activation marker CD44, on the other hand, was equivalent between these cell types (*Figure 4G*) suggesting that activation is intact. We hypothesized that reduced expression of Bcl-$_{XL}$ or ICOS might contribute to the reduced population of Tfh cells in *Cd28$^{flox/flox}$ Ox40$^{cre/+}$* mice. However, halving the amount of Bcl-$_{XL}$ (*Figure 4—figure supplement 1A–E*) or ICOS (*Figure 4—figure supplement 1F–H*) alone was not sufficient to cause increased Tfh cell death and reduced Tfh number during influenza A virus infection. Together, these data demonstrate that CD28 is required for the survival of the Tfh population and suggest that the loss of continued CD28 signaling affects the expression of multiple molecules in Tfh cells.

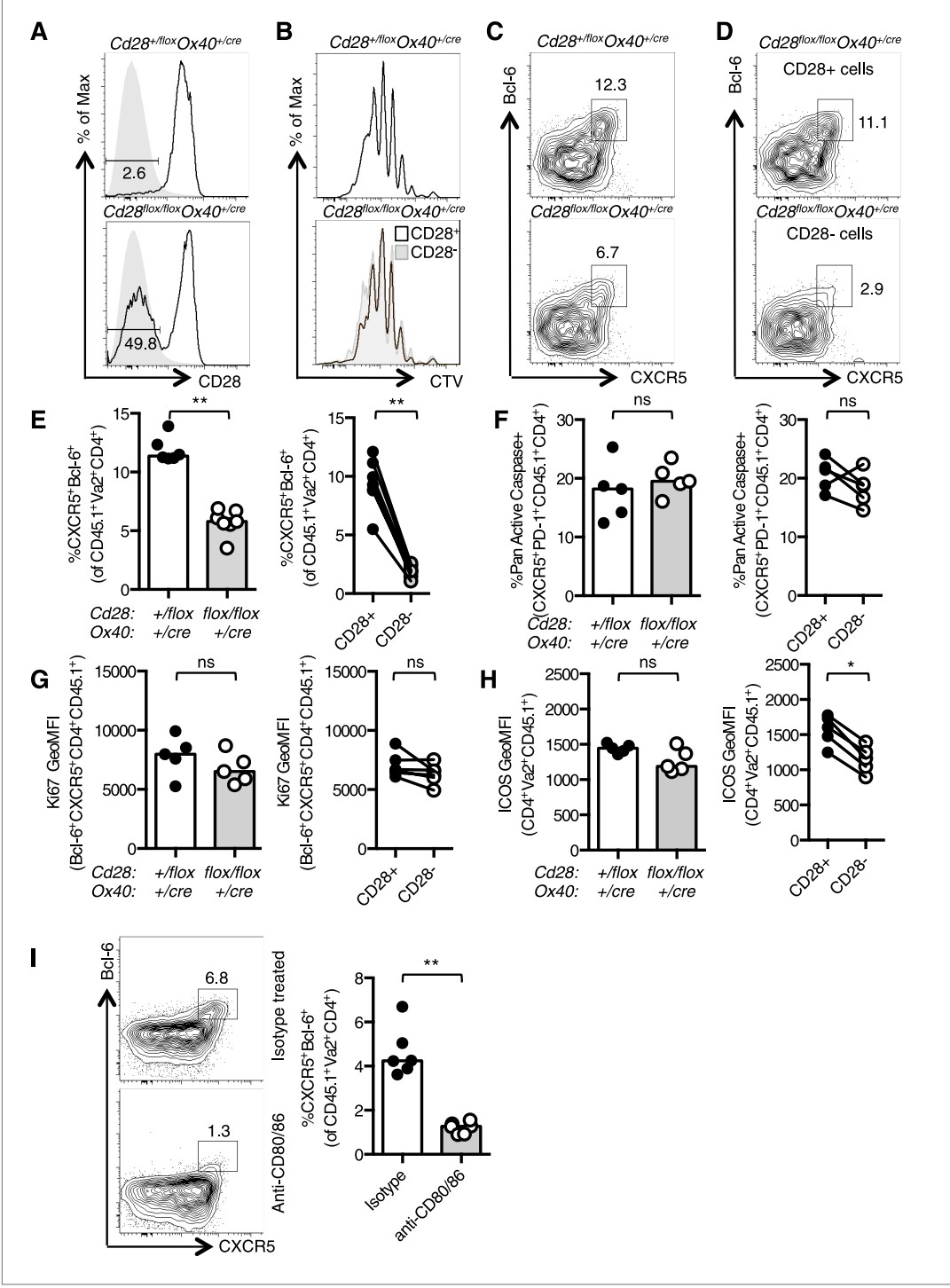

**Figure 3**. Tfh formation required CD28 signaling after T cell activation. 1 × 10⁵ OT-II T cells were and transferred into CD45.1 C57BL/6 hosts and immunized with OVA I.P. CD28 expression (**A**) on splenic CD4⁺Va2⁺CD45.2⁺ cells 3.5 days after immunization, gray filled histograms show isotype controls. Dilution of CTV (**B**) in CD28+ cells (open histograms) and CD28- cells (gray histogram). Bcl-6⁺CXCR5⁺CD4⁺Foxp3⁻ pre-Tfh cells in transferred OT-II *Cd28^(+/flox) Ox40^(cre/+)* and OT-II *Cd28^(flox/flox) Ox40^(cre/+)* cells (**C**) and in CD28+ and CD28- cells (**D**) from OT-II *Cd28^(flox/flox) Ox40^(cre/+)* cells. Proportion of Bcl-6⁺CXCR5⁺CD4⁺Foxp3⁻ pre-Tfh cells (**E**), percentage of Pan-Active Caspase⁺ pre-Tfh (**F**), expression of Ki-67 in pre-Tfh (**G**) and ICOS on pre-Tfh (**H**) from transferred OT-II *Cd28^(+/flox) Ox40^(cre/+)* and OT-II *Cd28^(flox/flox) Ox40^(cre/+)* cells (left graphs) and in CD28+ and CD28- cells from OT-II *Cd28^(flox/flox) Ox40^(cre/+)* cells (right graphs).
*Figure 3. Continued on next page*

*Figure 3. Continued*

Paired samples are connected with a line. (**I**) Pre-Tfh 4 days after transfer of $1 \times 10^5$ OT-II T cells into CD45.2 C57BL/6 hosts immunized with OVA I.P. followed by CD80/86 blocking antibodies I.V. 64 hr following immunization. Heights of the bars represent the median values. **A–H**, data are representative of four experiments with 5–7 mice per group. **I** is representative of three experiments with six mice per group. Ns = not significant, **p < 0.005, ***p < 0.005.

To confirm these findings in an independent experimental system, C57BL/6 mice were infected with influenza and treated with CD80/86 blocking antibodies on days 11 and 13 after infection (*Figure 5A*). 15 days post infection the percentage of Bcl-6$^+$Ki67$^+$B220$^+$ germinal center B cells was not altered between mice that received isotype control or CD80/86 blocking antibodies (*Figure 5B*), but the proportion of Tfh cells was reduced after anti-CD80/86 treatment (*Figure 5C*). There was an increase in Tfh cell apoptosis and decreased Bcl-$_{XL}$ and ICOS expression (*Figure 5D–F*), consistent with the results from our *Cd28$^{flox/flox}$ Ox40$^{cre/+}$* mice, however, in contrast with these results there was a decrease in Tfh cell proliferation (*Figure 5G*), suggesting that blocking CD28 and/or CTLA-4 on all T cells also impacts the on the proliferation of Tfh cell pool. Together, these data demonstrate that CD28 is required for maintenance of the Tfh population.

## Th1 differentiation requires continued CD28 signaling

Loss of CD28 after T cell activation impairs the Th1 response over the course of an infection because at day 12, but not at day 5, post infection there are fewer Th1 cells in *Cd28$^{flox/flox}$ Ox40$^{cre/+}$* mice compared with controls (*Figure 2E,F*). The proportion of CD28-negative Th1 cells in the lung remained constant between 5 and 14 days post influenza infection (*Figure 6A,B*), suggesting that, unlike Tfh, continued CD28 expression is dispensable for Th1 maintenance. Consistent with this, blocking CD80/86 at d11 and 13 post infection did not affect the proportion of Th1 cells in the medLN or lung (*Figure 6—figure supplement 1A,B*) 15 days post infection.

After infection there is expansion of influenza-specific T cells in the medLN, peaking at 6 days post infection, accompanied by substantial migration of antigen-specific effector CD4$^+$ T cells into the lung (*Roman et al., 2002*). As continued CD28 was dispensable for Th1 maintenance, we investigated whether CD28 signaling was important for Th1 expansion beyond day 5 post infection. A smaller proportion of IFNγ$^+$CD44$^{high}$CD4$^+$ medLN Th1 cells was detected in *Cd28$^{flox/flox}$ Ox40$^{cre/+}$* mice (*Figure 6C*) at 7 days post infection, just after the reported peak of CD4$^+$ T cell expansion (*Roman et al., 2002*). This reduction could not be accounted for solely by the failure of CD28-negative cells to differentiate into Th1 cells (*Figure 6D*), suggesting that the defect may not be due to cell-intrinsic CD28 signaling, but may be due to changes in factors that also affect surrounding cells, such as cytokines or chemokines. *Cd28$^{flox/flox}$ Ox40$^{cre/+}$* CD44$^{high}$CD4$^+$ cells showed normal apoptosis (*Figure 6E*), but a decrease in proliferation (*Figure 6F*) 7 days following infection. This defect in proliferation was not observed in Th1 cells 14 days post infection, suggesting that it is limited to the expansion phase of the Th1 response (*Figure 6—figure supplement 1C*).

As IL-2 is one of the key mitogenic cytokines involved in T cell expansion during influenza infection (*Sarawar and Doherty, 1994*), reduced IL-2 production may be responsible for the reduced proliferation of activated T cells in *Cd28$^{flox/flox}$ Ox40$^{cre/+}$* mice. Consistent with this, the proportion of IL-2 secreting cells in *Cd28$^{flox/flox}$ Ox40$^{cre/+}$* mice was decreased (*Figure 6G*), and likely contributes to reduced expansion of effector T cells in *Cd28$^{flox/flox}$ Ox40$^{cre/+}$* mice. Upregulation of CXCR3, a tissue homing receptor expressed on Th1 cells (*Beima et al., 2006*) was also impaired (*Figure 6H*) in *Cd28$^{flox/flox}$ Ox40$^{cre/+}$* mice, and the proportion of CXCR3$^+$CD62L$^{low}$CD4$^+$ cells was reduced (*Figure 6I*), suggesting that fewer activated Th1 cells have the capability to migrate to the lung. Indeed, 7 days following infection there were fewer Th1 cells in the lung in *Cd28$^{flox/flox}$ Ox40$^{cre/+}$* mice (*Figure 6J*). These data suggest that the loss of CD28 on activated T cells results in decreased IL-2 production and impaired acquisition of a migratory phenotype by activated CD4$^+$ T cells, resulting in fewer Th1 cells in the medLN and lung.

## Impaired CD4$^+$ effector T cell responses do not affect resolution of influenza infection

Primary intranasal infection with HKx31 results in high viral titres in the lung in the first 5 days of infection; at 7 days post infection, the virus has begun to be cleared and there is no detectable virus present at 11 days post infection (*Flynn et al., 1999*). *Cd28$^{flox/flox}$ Ox40$^{cre/+}$* mice had comparable levels of

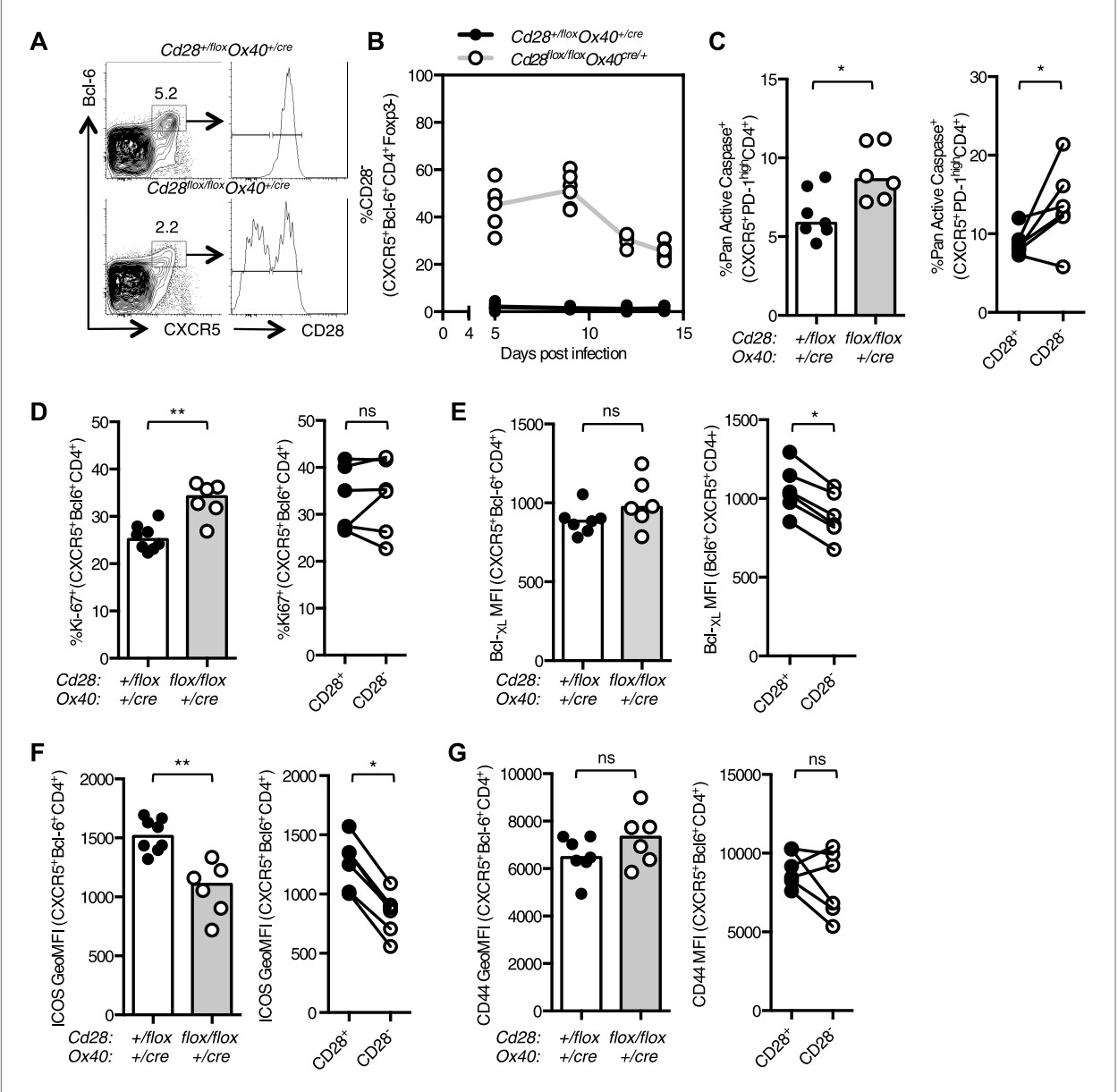

**Figure 4**. Maintained CD28 expression is required for the maintenance of Tfh cells during influenza infection. Cd28$^{flox/flox}$ Ox40$^{cre/+}$ mice and heterozygous controls were infected I.N. with 10$^4$ plaque-forming units of influenza A virus. The percentage of Bcl-6$^+$CXCR5$^+$CD4$^+$Foxp3$^-$ Tfh cells that lose CD28 expression was quantified at days 5, 9, 12, and 14 post infection in the medLN (**A** and **B**). The proportion of Pan Active caspase$^+$ Tfh cells (**C**), Ki-67$^+$ Tfh cells (**D**), the expression of Bcl-$_{XL}$ (**E**), ICOS (**F**) and CD44 (**G**) (%CD44$^+$ mean 92.7% for *Cd28$^{+/flox}$ Ox40$^{cre/+}$* and 89.1% for *Cd28$^{flox/flox}$ Ox40$^{cre/+}$*) on Tfh cells were enumerated at d14 post infection, comparing between *Cd28$^{flox/flox}$ Ox40$^{cre/+}$* mice and heterozygous controls (left graphs) and also between CD28+ and CD28- Tfh cells (right graphs) from *Cd28$^{flox/flox}$ Ox40$^{cre/+}$* mice. Paired samples from the same mouse are connected with a line and heights of the bars in bar graphs represent the median value. Data are representative of four independent experiments with 5–7 mice per group. Ns = not significant, *p < 0.05, **p < 0.005, ***p < 0.005.

The following figure supplement is available for figure 4:

**Figure supplement 1**. Reduced Bcl-$_{XL}$ or ICOS expression does not impair the Tfh or Th1 population during influenza infection.

Influenza M1 viral RNA in their lungs compared with heterozygous control mice at 7 days post infection (***Figure 7A***), suggesting that the loss of CD28 on effector CD4$^+$ T cells does not impair viral clearance. This is likely because CD4$^+$ T cells are not essential for clearance of primary HKx31 infection (***Tripp et al., 1995***; ***Belz et al., 2002***). Furthermore, we observed that influenza-specific CD8$^+$ T cells

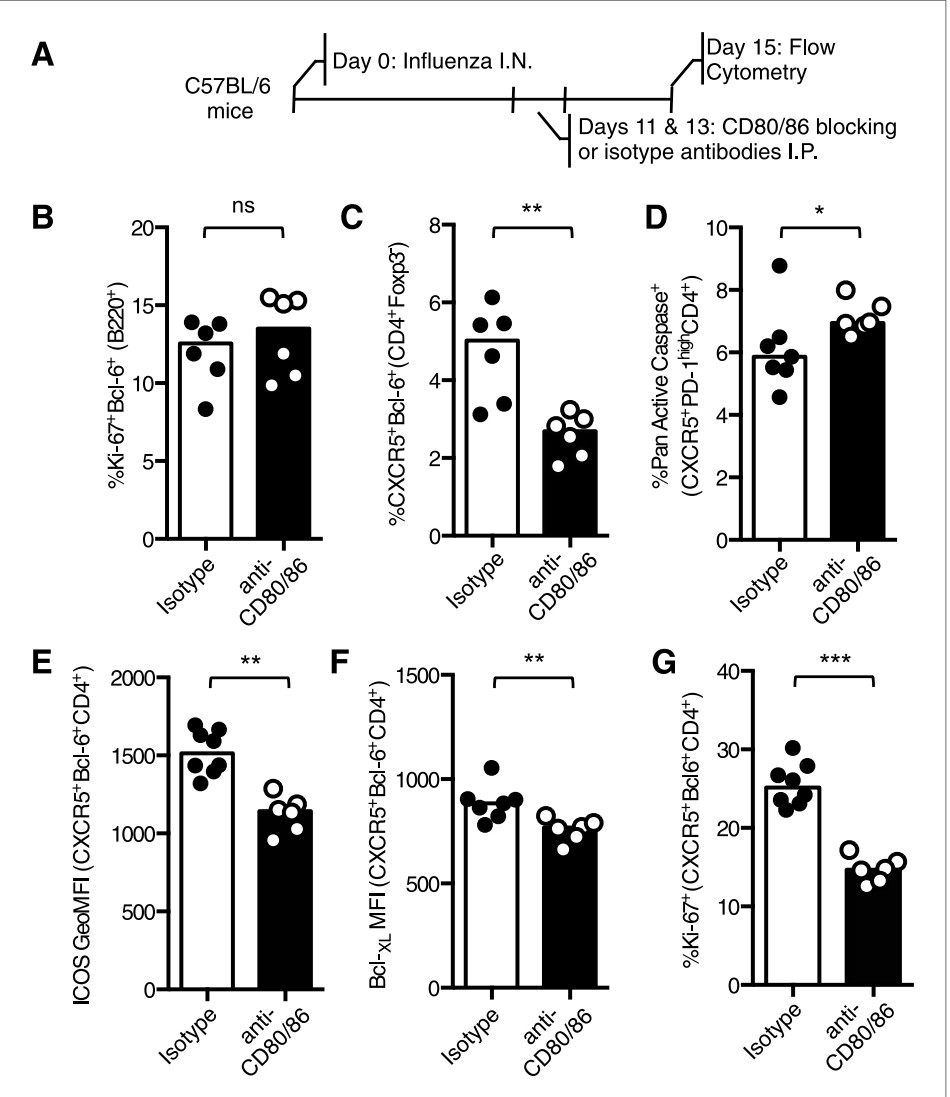

**Figure 5**. Blocking CD28 ligands reduced Tfh cells in an established infection. (**A**) C57BL/6 mice were infected I.N. with influenza A virus and administered CD80/86 blocking antibodies I.P. at days 11 and 13 post infection. The populations of medLN germinal center Bcl-6+Ki67+B220+ B cells (**B**) and medLN Bcl-6+CXCR5+CD4+Foxp3- Tfh cells (**C**) were measured by flow cytometry. The percentage of apoptotic pan active caspase+ Tfh cells (**D**), expression of ICOS (**E**) and Bcl-xL (**F**), and proportion of proliferating Ki67+ Tfh cells (**G**) on Tfh cells was assessed by flow cytometry. Heights of the bars represent the median values. Data are representative of three independent experiments with 6–7 mice per group. Ns = not significant, *p < 0.05, **p < 0.005, ***p < 0.005.

were present in comparable numbers in *Cd28*flox–flox *Ox40*cre/+ and control mice (**Figure 7B–D**) and may mediate viral clearance during primary infection.

## Functional differentiation of Foxp3+ Tregs requires CD28 signaling

Conditional ablation of CD28 in Foxp3+ regulatory T cells has demonstrated that there is an intrinsic role for CD28 in Treg function (**Zhang et al., 2013**). As OX40, and therefore Cre in our system, is expressed in Foxp3+ regulatory T cells (Tregs) in the thymus (**Klinger et al., 2009**), we assessed the proportion and phenotype of Tregs in *Cd28*flox/flox *Ox40*cre/+ mice. Splenic Tregs from uninfected *Cd28*flox/flox *Ox40*cre/+ mice had lost CD28 expression (**Figure 8A**), and there was a small reduction in the proportion of Foxp3+CD4+ cells (**Figure 8B**). Consistent with the previous report (**Zhang et al., 2013**), a smaller proportion of *Cd28*flox/flox *Ox40*cre/+ Tregs were proliferating (**Figure 8C**), and they had lower expression of the CD28-inducible inhibitory receptors PD-1 and CTLA-4 (**Figure 8D,E**). In mixed chimeras, Treg

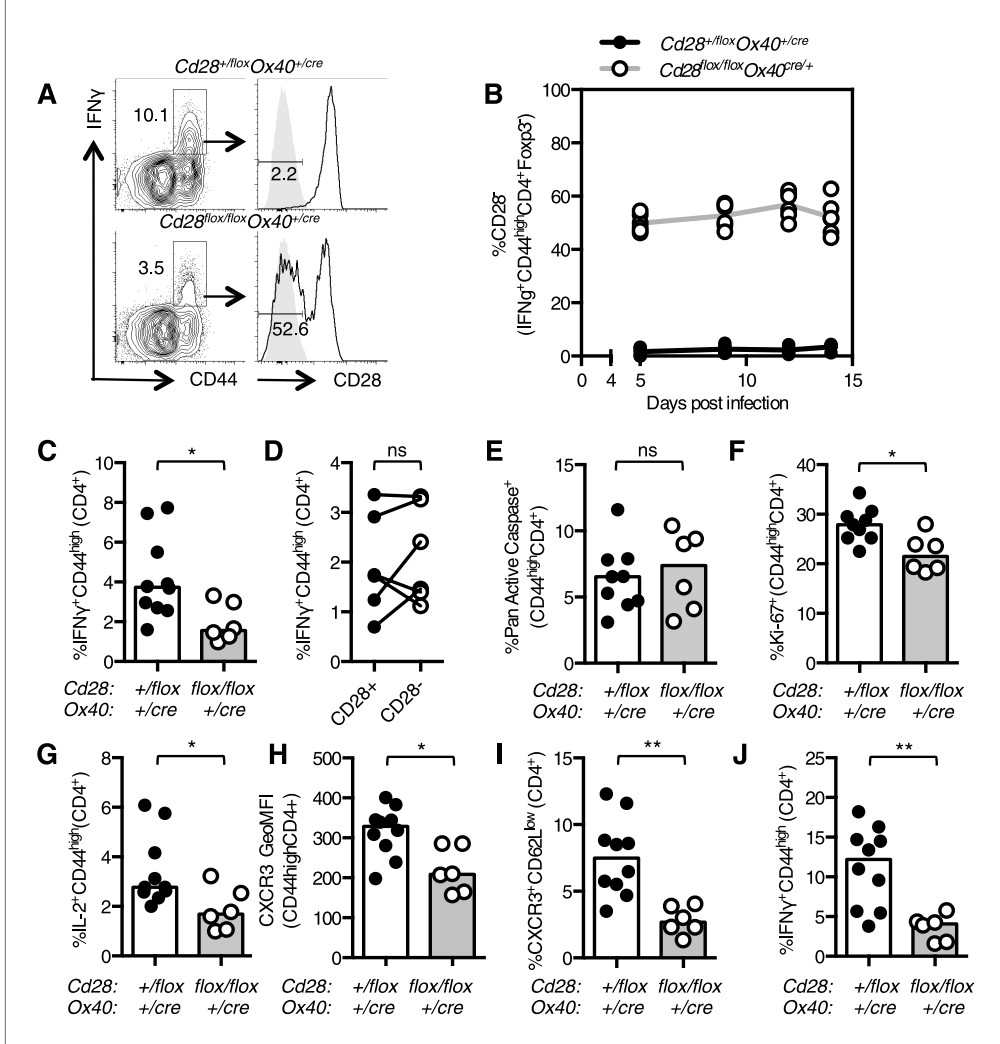

**Figure 6**. Th1 expansion requires maintained CD28 signaling. *Cd28flox/flox Ox40cre/+* mice and heterozygous controls were infected I.N. with 10⁴ plaque-forming units of influenza A virus. The percentage of IFNγ⁺CD44highCD4⁺ Th1 cells that lose CD28 expression was quantified at days 5, 9, 12, and 14 post infection in the lung (**A** and **B**). The proportion of IFNγ⁺CD44highCD4⁺ Th1 cells (**C**, **D**), apoptotic Pan Active Caspase⁺CD44highCD4⁺ cells (**E**), proliferating Ki-67⁺CD44highCD4⁺ cells (**F**), IL-2⁺CD44highCD4⁺ cells (**G**), expression of CXCR3 on CD44highCD4⁺ cells (**H**), the proportion of CXCR3⁺CD62LlowCD4⁺ (**I**) cells were assessed in the medLN 7 days post infection. The percentage of lung IFNγ⁺CD44highCD4⁺ Th1 cells (**J**) was also assessed at the same time point. In **D**, paired samples from the same mouse are connected with a line and heights of the bars in bar graphs represent the median value. Data are representative of three independent experiments with 5–10 mice per group. Ns = not significant, *p < 0.05, **p < 0.005, ***p < 0.005.

The following figure supplement is available for figure 6:

**Figure supplement 1**. Maintenance of lung Th1 cells is independent of CD28 expression.

cells derived from *Cd28flox/flox Ox40cre/+* bone marrow were outcompeted by control Tregs, suggesting the loss of CD28 affects Treg fitness (**Figure 8F**). As our previous data had shown that maintained CD28 signaling was required for both Th1 and Tfh cells, we asked whether CD28 was also required for their regulatory counterparts: T-bet⁺CXCR3⁺Foxp3⁺CD4⁺ Th1-type Tregs cells that form during type 1 inflammation (**Koch et al., 2009**) and Bcl-6⁺CXCR5⁺Foxp3⁺CD4⁺ T follicular regulatory cells that participate in the germinal center response (**Chung et al., 2011**; **Linterman et al., 2011**; **Wollenberg et al., 2011**). 12 days after influenza infection, the proportion of Th1-type Tregs cells was reduced in the medLN (**Figure 8G,H**) and lung (**Figure 8I**), likewise the proportion of T follicular regulatory cells in

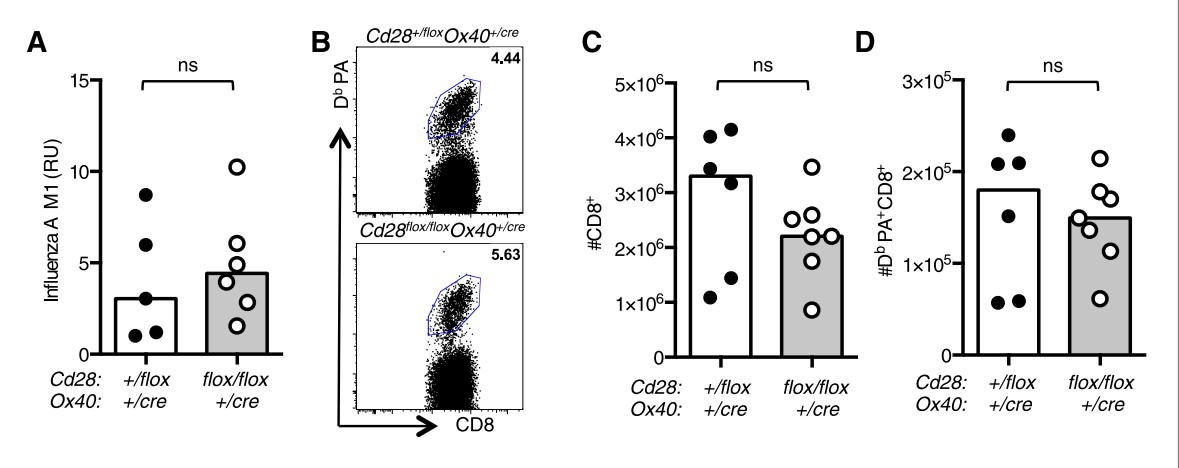

**Figure 7**. Impaired CD4[+] effector T cell responses do not affect resolution of influenza infection. (**A**) RNA encoding the influenza A virus M1 protein was determined in the lung 7 days post infection by qRT-PCR. Each dot represents the mean expression of three technical replicates per mouse. (**B**) Representative flow cytometric dot plots of CD8 cells binding the influenza epitope PA224-233 as identified by D[b] PA-APC dextramer 12 days after influenza A infection. Total CD8[+] cells (**C**) and D[b] PA dextramer binding CD8[+] cells (**D**) are enumerated in bar graphs. Each symbol represents one mouse and heights of the bars show the median values. Data are representative of two independent experiments with 5–6 mice per group. ns = not significant.

the medLN was reduced in *Cd28*[flox/flox] *Ox40*[cre/+] mice (**Figure 8J,K**), consistent with the lack of T follicular regulatory cells in CD28-deficient mice (**Linterman et al., 2011**; **Sage et al., 2013**). Together, our data suggest that CD28 signals are important for the functional differentiation of Foxp3[+] regulatory cells.

## Clearance of intestinal *Citrobacter rodentium* infection is defective in *Cd28*[flox/flox] *Ox40*[cre/+] mice

To determine if maintained CD28 expression is important for the immune response to bacterial infection, and not only viral infection, we infected *Cd28*[flox/flox] *Ox40*[cre/+] mice with *Citrobacter rodentium*, an enteric mucosal murine pathogen that elicits a robust humoral and cellular immune response. CD4[+] T cells play a central role in immunity to *C. rodentium*: CD4[+] T cells deficient for CD28, IFNγ, Interleukin 17 (IL-17), or the B cell helper molecule CD40L, are unable to support clearance of *C. rodentium* from the gut (**Bry et al., 2006**; **Ishigame et al., 2009**; **Shiomi et al., 2010**). After oral infection, heterozygous control mice had no detectable *C. rodentium* in their feces 24 days post infection, by contrast, *Cd28*[flox/flox] *Ox40*[cre/+] mice were still infected (**Figure 9A**). *Cd28*[flox/flox] *Ox40*[cre/+] mice had high bacterial burden in their cecum and colon 24 days post infection (**Figure 9B,C**), while the control mice had no detectable *C. rodentium* remaining in the gut (**Figure 9B,C**). However, *C. rodentium* had been cleared from the liver in *Cd28*[flox/flox] *Ox40*[cre/+] mice suggesting that the infection was not entirely uncontrolled (**Figure 9D**). We assessed the cellular immune response in the mesenteric lymph node 12 days post *C. rodentium* infection, when the bacterial burden is equivalent between the two groups, and observed a smaller germinal center response (**Figure 9E**), a decreased proportion of Tfh cells (**Figure 9F**) and Th1 cells (**Figure 9G**) in *Cd28*[flox/flox] *Ox40*[cre/+] mice. Furthermore, the population of IL-17[+]CD44[high]CD4[+] Th17 cells was reduced in the mesenteric lymph node of *Cd28*[flox/flox] *Ox40*[cre/+] mice 7 days post infection (**Figure 9H**). In addition to CD4[+] helper T cells, group 3 innate lymphoid cells (ILC3) have also been implicated in the clearance of *C. rodentium* infection (**Sonnenberg et al., 2011**). However, in *Cd28*[flox/flox] *Ox40*[cre/+] mice, there are comparable numbers of ILC3 cells in the colon compared to control mice 7 days post infection (**Figure 9I**), suggesting that an impairment in ILC3 number does not contribute to the impaired clearance of *C. rodentium* in *Cd28*[flox/flox] *Ox40*[cre/+] mice. These data demonstrate that maintained CD28 expression on CD4[+] T cells is required for full effector CD4[+] T cell populations and clearance of *C. rodentium*.

## Discussion

CD28 is essential for CD4[+] T cell priming and differentiation into effector T cell subsets, and here, we demonstrate a previously undescribed role for CD28 following T cell activation. We show that CD28

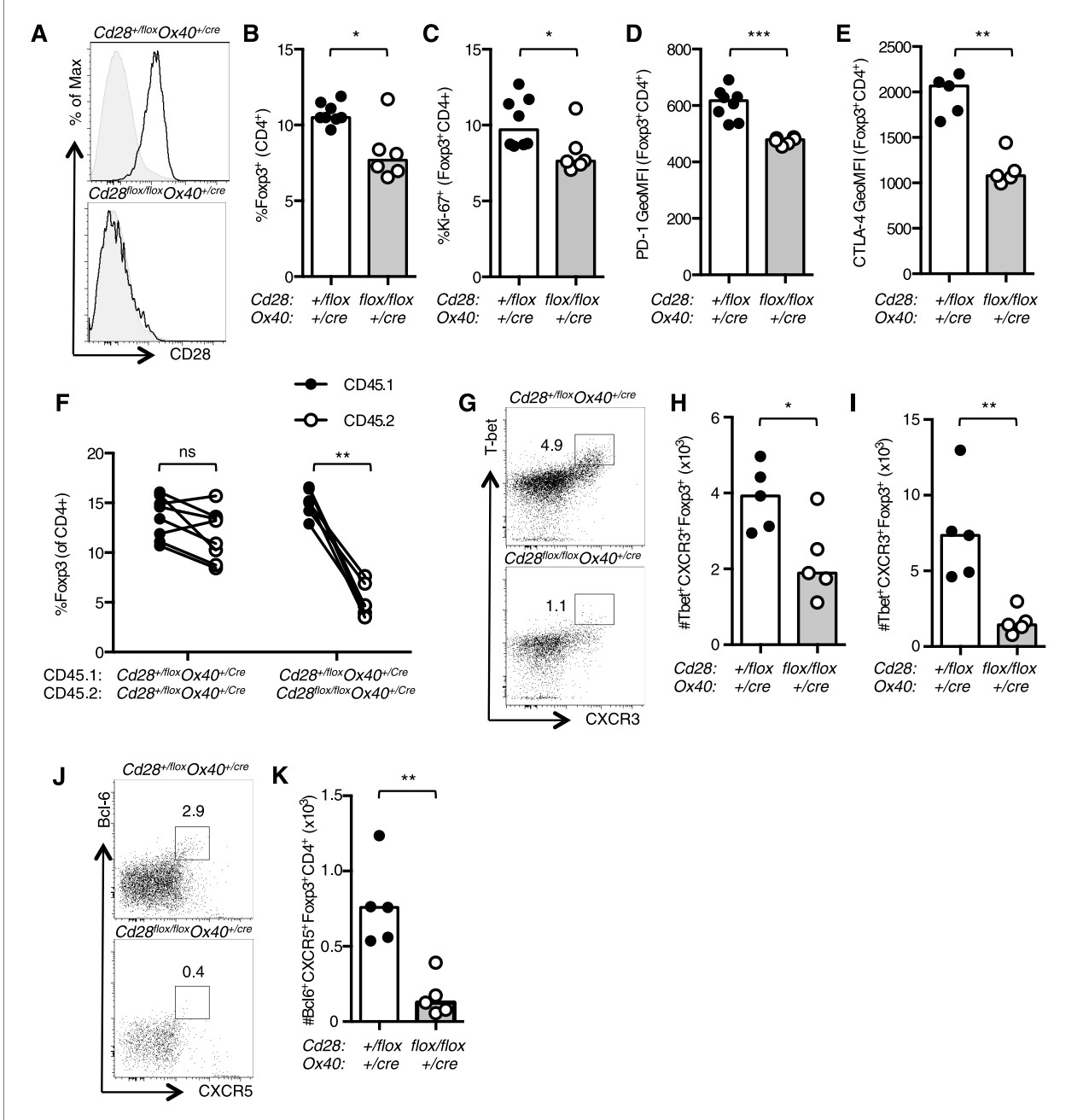

**Figure 8**. CD28 is required for Treg differentiation. Histograms of CD28 expression on Foxp3+CD4+ splenocytes (**A**) from *Cd28^flox/flox^ Ox40^cre/+^* mice and heterozygous controls. Proportion of Foxp3+ Tregs within the CD4+ splenic T cell population (**B**). Percentage of proliferating Ki-67+ Tregs (**C**) and expression of PD-1 (**D**) and CTLA-4 (**E**) on splenic Tregs. (**F**) Regulatory T cells from mixed bone marrow chimeras of specified genotypes 12 days post influenza A virus infection. (**G–K**) *Cd28^flox/flox^ Ox40^cre/+^* mice and heterozygous controls were infected I.N. with 10^4^ plaque-forming units of influenza A virus and 14 days following influenza infection the percentage and number of Tbet+CXCR3+Foxp3+CD4+ cells in the medLN (**G** and **H**), lung (**I**), and Bcl-6+CXCR5+Foxp3+CD4+ Tfr cells in the medLN (**J** and **K**) were assessed by flow cytometry. The heights of the bars represent median values. Data are representative of three independent experiments with 5–8 mice per group. Ns = not significant, *p < 0.05, **p < 0.005, ***p < 0.005.

expression is required after T cell priming for the expansion of Th1 cells and for the differentiation and maintenance of Tfh cells. Maintained CD28 expression is also required for clearance of the entero-pathic bacteria *C. rodentium* after oral infection. Our data demonstrate that even though maintained CD28 expression is required for both Th1 and Tfh cells, the downstream effects of CD28 are different for each of these cell types.

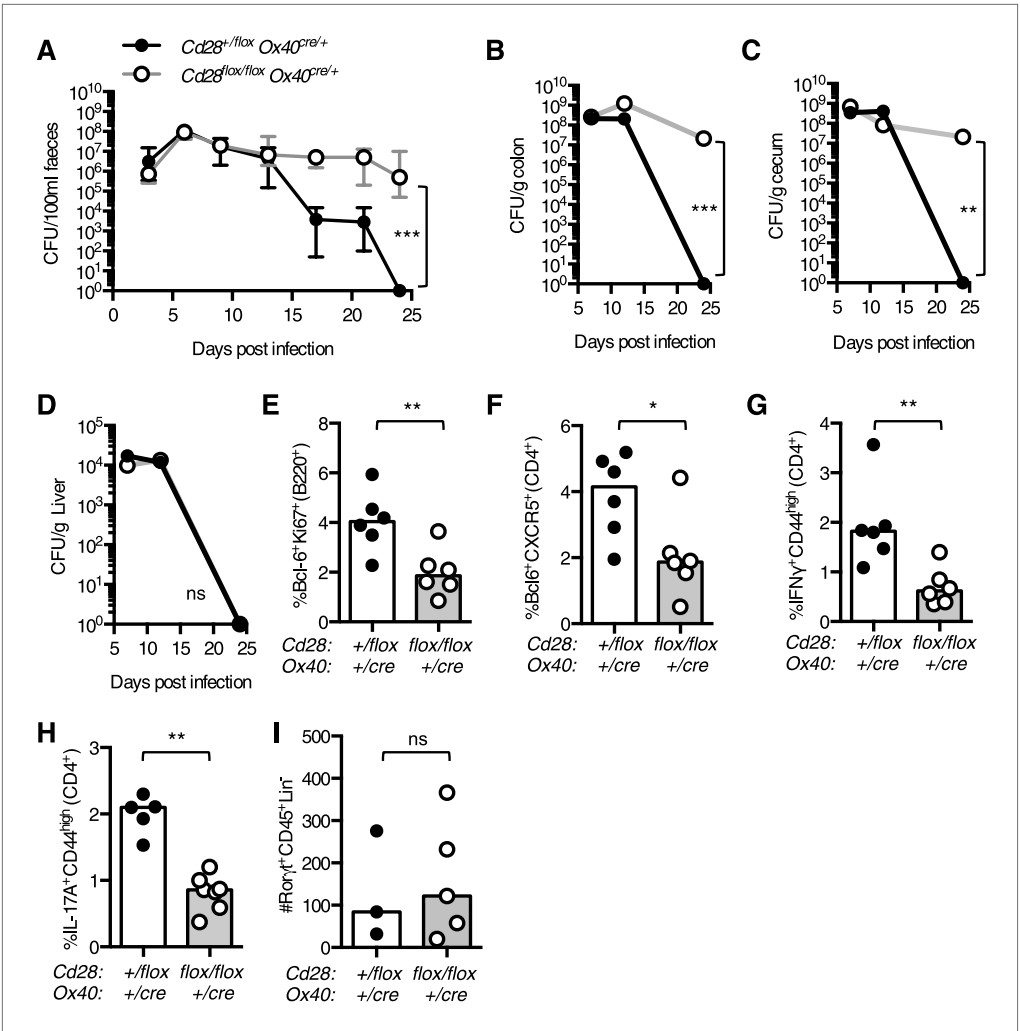

**Figure 9**. Maintained CD28 expression is required for clearance of *C. rodentium*. *Cd28*flox/flox *Ox40*cre/+ mice and heterozygous controls were infected orally with *C. rodentium* and CFU/100 ml of feces was determined at 3–4 day intervals for 24 days (**A**). Bacterial load in the colon (**B**), cecum (**C**), and liver (**D**) was enumerated 7, 12 and 24 days post infection. 12 days post infection, the percentage of Bcl-6+Ki-67+B220+ germinal center B cells (**E**), Bcl-6+CXCR5+CD4+Foxp3− Tfh cells (**F**) and IFNγ+CD44high CD4+ Th1 cells (**G**) was assessed in the mesenteric lymph node. The percentage of IL-17A+CD44high CD4+ Th17 cells was assessed in the mesenteric lymph node (**H**), and the number of Rorγt+CD45+Lin− ILC3 cells was assessed in the colon (**I**) 7 days post infection. The heights of the bars represent median values. In **A**, the error bars show the range. In **B–D**, the error bars show SEM. Ns = not significant, *p < 0.05, **p < 0.005, ***p < 0.005.

The impaired numbers of Th1 at day 7 but not earlier in the response is likely to be due to reduced proliferation of activated CD4+ T cells, which is IL-2-driven at this time in the response (*Sarawar and Doherty, 1994*). This observation is supported by previous in vitro experiments that show activated Th1 cells, but not Th2 cells, require CD80/86 for IL-2 production and proliferation in vitro (*Schweitzer and Sharpe, 1998*). In contrast, proliferation of Tfh-precursors is independent of IL-2 (*Choi et al., 2011*), and consistent with this, we have shown that Tfh-precursors from *Cd28*flox/flox *Ox40*cre/+ mice have comparable proliferation rates to controls. However, Tfh-precursors from *Cd28*flox/flox *Ox40*cre/+ mice are reduced in number, and this is most pronounced in cells that have lost CD28. This occurs at a time in the response where B cells provide a second round of antigen presentation to Tfh-precursors, allowing for stabilization of Bcl-6 expression and full differentiation into Tfh cells (*Baumjohann et al., 2011*; *Choi et al., 2011*), suggesting that CD28:CD80/86 interactions are important for Tfh differentiation during early T:B interactions. We identified that CD28 is also required for maintenance of the Tfh

population in established germinal centers, by preventing apoptosis of these cells. This result was somewhat surprising in light of the work of *Walker et al. (2003)* who used a CTLA-4-Ig transgenic mouse to show that blocking both CTLA-4 and CD28 during an established GC increased the size of the response. This, taken together with our data and the report that CTLA-4-deficient mice form spontaneous GC (*Wing and Sakaguchi, 2014*), suggests that CTLA-4 plays a more dominant role in suppressing the GC response than the role for CD28 in supporting the response via Tfh maintenance. Our results are consistent with those of *Good-Jacobson et al. (2012)* who reported that mice lacking CD80 on B cells have fewer Tfh cells due to an increased frequency of apoptosis. However, because CD80 is a ligand for CD28, CTLA-4, and PD-L1, this study could not discriminate which binding partner mediated this effect (*Keir et al., 2008*). Our study, together with this work from Good-Jacobson et al., suggests that interaction of CD28 with CD80 is important for maintaining the Tfh population, whereas CD86 appears to have a more dominant role in the formation of Tfh cells (*Salek-Ardakani et al., 2011*). Furthermore, it has recently been shown that IL-21, a cytokine produced by Tfh cells, can induce CD86 upregulation on B cells (*Attridge et al., 2014*). This suggests that Tfh cells can stimulate B cells to express ligands required for their own formation, and that the dialogue between Tfh and B cells is reciprocal, and may form a positive feedback loop that supports the germinal center response. Our data suggest that the role for CD80/86 in Tfh biology occurs through interactions of CD28, rather than CTLA-4. Together, this indicates that the interactions between CD28 and CD80 may be important for maintaining the Tfh population, and that the dominant role of CD86 in promoting Tfh formation is likely to be driven by interactions with CD28, rather than CTLA4.

The results presented here demonstrate that continued CD28 signaling is required following T cell activation for an effective primary CD4$^+$ T cell response to infection. In addition, there is evidence that CD28 is required for the initiation of CD4$^+$ T cell memory responses. Recent work from *Ndlovu et al. (2014)* used an inducible deletion system to remove CD28 prior to secondary infection with *Nippostronglus brasiliensis,* a nematode that induces a Th2-skewed response. The authors show that deletion of CD28, after clearance of the first infection and prior to re-infection, resulted in fewer Th2 cells and Tfh cells forming in response to secondary infection. This demonstrates that CD28 signaling is required for the initiation of the CD4$^+$ memory T cell response, in addition to being required throughout the primary immune response to infection.

CD28 signaling is essential for the formation of regulatory T cells and for their suppressive function in the periphery (*Zhang et al., 2013*). We have shown here that CD28 is also required for the presence of the Tfr and Tbet$^+$ Treg subsets. The lack of Tbet$^+$ Treg and Tfr cells in *Cd28$^{flox/flox}$ Ox40$^{cre/+}$* mice suggests that the germinal center and the Th1 response are not under the normal Treg-mediated control (*Koch et al., 2009*; *Chung et al., 2011*; *Linterman et al., 2011*), and this would likely result in a relative expansion of these cells. Mixed bone marrow chimera experiments, in which wild-type Tregs are present from control bone marrow show Tfh and Th1 from *Cd28$^{flox/flox}$ Ox40$^{cre/+}$* are almost absent. This may suggest that, in the presence of an intact Treg compartment, effector *Cd28$^{flox/flox}$ Ox40$^{cre/+}$* T cells are less likely to expand. However, in this mixed chimera system, we cannot exclude the equally likely possibility that CD28$^-$ T cells are simply less fit than CD28$^+$ T cells and are outcompeted.

CD28 signaling initiates a specific transcriptional program within T cells, modulating the expression of hundreds of genes (*Martinez-Llordella et al., 2013*). It is likely that during cognate interactions of Tfh with B cells—both prior to and during the germinal center response—CD28 ligation triggers a number of changes that support Tfh differentiation and maintenance. We identified reduced expression of ICOS and Bcl-$_{XL}$ in CD28$^-$ Tfh cells, and reduced PD-1 and CTLA-4 in Foxp3$^+$ Tregs as examples of these changes. However, we show that reducing expression of either ICOS or Bcl-$_{XL}$ alone was not sufficient to recapitulate the reduction in Tfh cell number seen in *Cd28$^{flox/flox}$ Ox40$^{cre/+}$* mice. This suggests that, like its role during T cell priming, CD28 signaling in activated T cells changes the expression of a large number of molecules, which together contribute to a reduced fitness of activated CD4$^+$ T cells that have lost CD28.

Our data demonstrate that the loss of CD28 on activated T cells can impair the immune response to foreign antigen; this has implications for the accumulation of CD28-negative CD4$^+$ T cells with age, during HIV infection, and in multiple immune disorders (*Weng et al., 2009*; *Aberg, 2012*; *Broux et al., 2012*). Both ageing and HIV infection are associated with impaired immune responses to foreign antigens (*Cubas et al., 2013*; *Linterman, 2014*), and part of this might be due to an increasing proportion of CD28-negative CD4$^+$ T cells that are not able to respond appropriately to foreign antigen. Furthermore, blocking the CD28 pathway is used to treat a number of autoimmune diseases and

to prevent allograft rejection, conversely this pathway can be stimulated to increase T cell immunity (*Salomon and Bluestone, 2001*; *Scalapino and Daikh, 2008*). Our data reveal that CD28 is not only important for T cell activation, but that it also plays important roles in the ongoing immune response. Therapeutic targeting of CD28 would therefore be expected to effectively dampen established T cell responses as well as interfering with priming. This is especially important for the treatment of autoimmunity, where the initial priming event would have already occurred, and explains how CD28 intervention is effective at controlling ongoing autoimmune disease. Furthermore, our results suggest that the loss of CD28 on activated T cells with age, or during chronic immune stimulation, may reduce the number of T cells that are able to respond to an infection, thereby functionally restricting the immune response and resulting in relative immunodeficiency, as are well described in ageing and in HIV infection. Taken together, this illustrates that CD28 is a critical co-stimulatory molecule throughout the effector phase of the response to infection, which has significant implications for applied immunology.

## Materials and methods

### Mice

$Cd28^{flox/flox}$ embryonic stem cells were obtained from the EUCOMM Consortium and used to generate chimeric animals from which the $Cd28^{flox/flox}$ strain was derived. C57BL/6, C57BL/6 $Rag2^{-/-}$, $Cd28^{flox/flox}Ox40^{cre/+}$, $Cd28^{flox/+}Ox40^{cre/+}$, CD45.1 $Cd28^{flox/+}Ox40^{cre/+}$, $Cd28^{+/flox}Ox40^{cre/+}ROSA^{stop-flox-RFP}$, $Cd28^{flox/flox}Ox40^{cre/+}ROSA^{stop-flox-RFP}$, OT-II $Cd28^{flox/flox}Ox40^{cre/+}$, OT-II $Cd28^{+/flox}Ox40^{cre/+}$ and $Icos^{+/-}$ mice were housed under specific-pathogen free conditions (Central Biomedical Services, University of Cambridge, UK).

### Infections and immunizations

To generate thymus-dependent germinal center responses, 10-week-old mice were immunized intraperitoneally with $2 \times 10^9$ sheep red blood cells (TCS bioscience, UK) or 10 µg of OVA (Sigma, UK) emulsified in Imject Alum (Thermo Scientific, UK). In adoptive transfer experiments, total splenocytes containing $1 \times 10^5$ Vα2⁺CD4⁺ OT-II cells were labeled with 5 µM Cell Trace Violet (Invitrogen, UK) and transferred by I.V. tail vein injection into recipient mice. For Influenza A infection, mice were infected with $10^4$ plaque-forming units of influenza A/HK/x31 virus (H3N2) I.N. under inhalation anesthesia with isoflurane. *C. rodentium* ICC180 inocula were prepared by culturing bacteria overnight at 37°C in 100 ml of Luria Bertani (LB) broth supplemented with nalidixic acid (100 µg/ml). Cultures were harvested by centrifugation and resuspended in a 1:10 volume of Dulbecco's phosphate-buffered saline (Sigma, UK). Mice were orally inoculated with 200 µl of the bacterial suspension by gavage needle.

### Measurement of *C. rodentium* burden

The number of viable bacteria in the feces and organs was determined by viable count on LB agar containing nalidixic acid (100 µg/ml).

### Cell isolation

Single cell suspensions were prepared from mouse lymphoid tissues by sieving and gentle pipetting through Falcon 70-µm nylon mesh filters (Becton Dickson, UK). Lung lymphocytes were isolated by finely mincing the lung tissues and digesting with 2 mg/ml Collagenase (Sigma, UK) and 0.2 mg/ml DNase I (Roche, UK) at 37°C for 30 min, followed by sieving and gentle pipetting through Falcon 70-µm nylon mesh filters (Becton Dickson, UK). Colon ILCs were isolated by longitudinal opening of the colon, then 0.5-cm fragments were shaken for 30 min at 37°C in PBS supplemented with 10% FCS, 10 mM EDTA, 20 mM HEPES, 1 mM sodium pyruvate, and 10 µg/ml Polymyxin B. The IEL fraction is then discarded and the colonic lamina propria lymphoid cells isolated via a further digest of the fragments with 1 mg/ml Collagenase D and 0.1 mg/ml DNAase I for an additional 45 min at 37°C.

### Antibodies for flow cytometry

Antibodies for flow cytometry were from eBioscience (UK) except where otherwise indicated: anti-Vα2 (B20.1), anti-DO11.10 TCR (KJ1-26), anti-CD28 (37.51), anti-IL-2 (JES6-5h4), anti-CD4 (GK1.5), anti-B220 (RA3-6B2), anti-ICOS (C378.4A; BioLegend, UK), anti-CD44 (IM7; BioLegend), anti-Ki67 (SolA15), anti-Bcl-6 (K112-91; Becton Dickinson, UK), anti-CXCR5 (2G8; Becton Dickinson), anti-IFNγ (XMG1.2), anti-CD45.1 (A20; BioLegend), anti-CD45.1 (104), anti-CXCR3 (CXCR3-173; BioLegend),

anti-CD62L (MEL-14), anti-Bcl-$_{XL}$ (7B2.5; AbCam), anti-PD-1 (29F.1A12; BioLegend), anti-CD8 (53-6.7), anti-IL-17A (eBio17B7), anti-Rorγt (B2D), anti-Tbet (eBio4B10), and anti-CD45 (30-F11). Active Caspases were detected by binding of the inhibitor VAD-FMK conjugated to sulfo-rhodamine (AbCam).

## Determination of influenza virus viral loads

Lung RNA was extracted using an RNeasy Micro Kit (QIAGEN, UK) as per the manufacturer's instructions. cDNA synthesis and real-time PCR were performed as described previously (*Borowski et al., 2007*). Briefly, cDNA was synthesized using both a specific primer (5′-TCTAACCGAGGTCGAAACGTA-3′) and random hexamers. Real-time assays were performed in triplicate with 5 μl of cDNA, 12.5 μl of 2 × TaqMan Universal PCR Master Mix (Applied Biosystems, UK), 900 nM influenza A virus sense primer (5′-AAGACCAATCCTGTCACCTCTGA-3′), 900 nM influenza A virus antisense primer (5′-CAAAGCGTCTACGCTGCAGTCC-3′), and 200 nM influenza A virus probe (FAM-5′-TTTGTGTTC ACGCTCACCGT-3′-TAMRA). All primers were specific for the influenza A virus matrix protein.

## Detection of influenza-specific CD8$^+$ T cells

Single cell suspensions were stained with 2 μl D$^b$PA-APC dextramer (Immudex, Denmark) specific for PA224–233 (sequence SSLENFRAYV) for 30 min at room temperature prior to staining for other cell surface markers at 4°C.

## Bone marrow chimeras

Recipient mice were sub-lethally irradiated with 1000 Rad and reconstituted via intravenous injection with 2 × 10$^6$ donor bone marrow cells. Chimeric mice were infected with influenza A virus 8 weeks after reconstitution.

## Immunohistochemistry

To visualize the germinal center response to SRBC immunization, spleen samples were fixed for 2 hr in 4% paraformaldehyde (PFA) on ice, incubated in six changes of sucrose buffer overnight, and embedded in Tissue-Tek OCT compound (Sakura Finetek, The Netherlands). Follicular B cells were stained with anti-mouse IgD Alexa Fluor 647 (11-26c.2a; BioLegend) and germinal center cells were detected with anti-mouse Bcl-6 FITC (N-3, Santa Cruz) followed by donkey anti-rabbit FITC (Jackson, UK) and then Alexa 488-conjugated goat anti-FITC (Invitrogen, UK). Stained sections were mounted in Fluoromount-g (Southern Biotech, Alabama 35260, USA) and analyzed with a confocal laser-scanning microscope (Zeiss LSM 710) using a 40x objective.

## Statistical analysis

Single comparisons were analyzed using the non-parametric Mann–Whitney U-test. Paired comparisons were analyzed using a Wilcoxon matched-pairs signed rank test. All statistical analyses were carried out with GraphPad Prism v6.

## Acknowledgements

We thank Adrian Liston and Klaus Okkenhaug for critical review of this manuscript, Andrew McKenzie and Jennifer Walker for provision of ICOS heterozygous mice and Daniel Gray for providing Bcl-$_{XL}$ ERT2cre and control bone marrow. This work was funded by a Wellcome Trust Program Grant (083650/Z/07/Z) to KGCS. The work was also supported by an NHMRC Overseas Biomedical Fellowship (to MAL), a Lister Prize Fellowship (to KGCS), by the National Institute of Health Research Cambridge Biomedical Research Centre and the Wellcome Trust grant number 098051 to the Sanger Institute.

## Additional information

### Funding

| Funder | Grant reference number | Author |
| --- | --- | --- |
| Wellcome Trust | Project Grant, 083650/Z/07/Z | Kenneth GC Smith |
| Lister Institute of Preventive Medicine | Lister Prize Fellowship | Kenneth GC Smith |

| Funder | Grant reference number | Author |
|---|---|---|
| National Health and Medical Research Council | Career Development Award, 596868 | Michelle A Linterman |
| Wellcome Trust | 098051 | Gordon Dougan |

The funders had no role in study design, data collection and interpretation, or the decision to submit the work for publication.

## Author contributions

MAL, Conception and design, Acquisition of data, Analysis and interpretation of data, Drafting or revising the article; AED, Acquisition of data, Analysis and interpretation of data, Drafting or revising the article; DPD, LK, CF, Acquisition of data, Analysis and interpretation of data; IZ, GD, Acquisition of data, Contributed unpublished essential data or reagents; MV, Analysis and interpretation of data, Contributed unpublished essential data or reagents; SC, Acquisition of data, Analysis and interpretation of data, Contributed unpublished essential data or reagents; ME, KGCS, Analysis and interpretation of data, Drafting or revising the article

## Author ORCIDs

Marc Veldhoen, http://orcid.org/0000-0002-1478-9562

## Ethics

Animal experimentation: All experiments were performed according to the regulations of the UK Home Office Scientific Procedures Act (1986) under the UK Home Office license PPL 80/2438, or PPL 80/2596.

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
