## [Decision Letter]

Thank you for sending your work entitled “CD28 expression is required after T cell priming for helper T cell responses and protective immunity to infection” for consideration at *eLife*. Your article has been evaluated by Tadatsugu Taniguchi (Senior editor), a Reviewing editor, and 3 reviewers, all of whom are members of our Board of Reviewing Editors.

The Reviewing editor and the other two reviewers discussed their comments before we reached this decision, and the Reviewing editor has assembled the following comments to help you prepare a revised submission.

In this manuscript Linterman et al. investigate the consequences of post priming loss of CD28 expression using a CD28 flox OX40cre system. Using this system they demonstrate that loss of CD28 has clear consequences for Tfh and Th1 function. Tfh displayed a need for CD28 signalling to progress from pre-Tfh to full Tfh, probably due to a loss of CD28 signalling via CD86 expressed by cognate B-cells, and a further requirement for CD28 to maintain the cells. Th1 cells on the other hand appear to require CD28 for their post-priming expansion but not their maintenance.

The paper is technically sound, well written and addresses an important question. This is an important area since CD28 signalling is a critical mechanism of T-cell activation and treatments targeting this pathway are already in clinical usage. It might be argued, however, that given the current understanding of CD28 signalling and the functional effects of blocking its ligands, the results are novel but not necessarily surprising. The authors should also reference and discuss recent studies using a CD28-inducible deletion during helminth infection that showed a requirement for CD28 for efficient Th2 immunity following re-infection (40).

Specific comments:

1) To provide further mechanistic insight the authors should perform bone marrow chimera experiments to shed light on which functions are cell intrinsic. This would be particularly interesting in the flu system. This will address possible effects of deletion of CD28 on regulatory T cells some of which express high levels of OX-40 under homeostatic conditions.

2) The point is made that manipulating CD80/86 cannot be used to directly infer information about CD28 signalling since they also bind to CTLA-4. While this is formally true given the high levels of IFNg and antibody production seen in CTLA-4 deficient mice, it does not seem likely that CTLA-4 enhances Tfh, Th1 or Germinal center formation.

3) There is some redundancy between Figures, especially with data from different time points of the same/repeated experiments. For example Figure 2 contains data concerning TH1 formation at days 5 and 12 while Figure 7 shows very similar data from day 7. Equally, Figure 7 shows caspase and Ki-67 data from day 7 while Figure 8 shows the same data at day 14. As far as I can tell, the methodology from these experiments is the same so, for clarity, it may be better to break up Figure 7 and combine it with the corresponding elements of Figures 2 and 8 to prevent the same data from different time points of the same experiments being presented in different figures.

4) Data showing the effects on bacterial clearance is shown in Figure 10. Many of the earlier figures demonstrate that Th1 and Tfh are impaired in influenza infection but the effects of this on the resolution of infection are not shown. Does the CD28flox/flox OX40cre phenotype affect the natural history of influenza infection in these mice?

5) Studies in the Citrobacter model would benefit from a kinetic analysis of bacterial growth allowing assessment of the impact on early innate and later adaptive responses. It will be particularly important to look at ILC3 and Th17 cells responses as these are important components of host protective immunity to this enteric pathogen.

---

## [Author Response]

*The paper is technically sound, well written and addresses an important question. This is an important area since CD28 signalling is a critical mechanism of T-cell activation and treatments targeting this pathway are already in clinical usage. It might be argued, however, that given the current understanding of CD28 signalling and the functional effects of blocking its ligands, the results are novel but not necessarily surprising. The authors should also reference and discuss recent studies using a CD28 inducible deletion during helminth infection that showed a requirement for CD28 for efficient Th2 immunity following re-infection (*[40]*)*.

This manuscript has now been included in the Discussion section.

Specific comments:

*1) To provide further mechanistic insight the authors should perform bone marrow chimera experiments to shed light on which functions are cell intrinsic. This would be particularly interesting in the flu system. This will address possible effects of deletion of CD28 on regulatory T cells some of which express high levels of OX-40 under homeostatic conditions*.

We agree that mixed chimera experiments can provide additional experimental insight. To this end we generated CD45.1 *Cd28*^*+/flox*^
*Ox40*^*cre/+*^ : CD45.2 *Cd28*^*flox/flox*^
*Ox40*^*cre/+*^ mixed chimeras, and control chimeras with CD45.1 *Cd28*^*+/flox*^
*Ox40*^*cre/+*^ : CD45.2 *Cd28*^*+/flox*^
*Ox40*^*cre/+*^ bone marrow and infected them with influenza, adding the data as a figure supplement to Figure 2. These chimeras show that Tfh and Th1 cells from *Cd28*^*flox/flox*^
*Ox40*^*cre/+*^ bone marrow represent a very small proportion of total effector cells. This is discussed in the Results section. In addition, we have also added data to Figure 8 (new panel F) showing that Foxp3^+^ Tregs cells from *Cd28*^*flox/flox*^
*Ox40*^*cre/+*^ bone marrow are underrepresented in mixed bone marrow chimeras, and the Results section has been amended to include this. These results are consistent with a cell-intrinsic role for CD28 on both CD4^+^ effector and regulatory T cells. This has also been expanded upon in the Discussion section.

*2) The point is made that manipulating CD80/86 cannot be used to directly infer information about CD28 signalling since they also bind to CTLA-4. While this is formally true given the high levels of IFNg and antibody production seen in CTLA-4 deficient mice, it does not seem likely that CTLA-4 enhances Tfh, Th1 or Germinal center formation*.

We agree with the reviewers on this point; it is clear that CD80/86 blocking experiments have yielded seminal insights into CD28 and CTLA-4 biology. We have altered a paragraph in the Introduction to emphasise this. However, we feel that it remains relevant to discuss a potential role for CTLA-4 in the manuscript, particularly as CD28 and CTLA-4 have opposing roles in T cell biology and it is possible that blocking CD80/86 may mask effects of blocking CD28 signalling if CTLA-4 was to play a dominant role. For example, Walker L.S.K. et al (JI 2003) use an elegant CTLA-4 transgenic mouse to block CD80/86 after GC induction, resulting in a larger GC response which is the opposite of what we observe in our *Cd28*^*flox/flox*^
*Ox40*^*cre/+*^ mice in which CD28 alone is targeted. This is likely to be because in the absence of both CD28 and CTLA-4 ligation, the effects of loss of CTLA-4 are more prominent. This has been elaborated on in the Discussion section.

*3) There is some redundancy between Figures, especially with data from different time points of the same/repeated experiments. For example*
Figure 2
*contains data concerning TH1 formation at days 5 and 12 while*
Figure 7
*shows very similar data from day 7. Equally,*
Figure 7
*shows caspase and Ki-67 data from day 7 while*
Figure 8
*shows the same data at day 14. As far as I can tell, the methodology from these experiments is the same so, for clarity, it may be better to break up*
Figure 7
*and combine it with the corresponding elements of*
Figures 2 and 8
*to prevent the same data from different time points of the same experiments being presented in different figures*.

We have pooled the key data from the original Figures 7 and 8 into Figure 6 to minimise redundancy, as suggested.

4) Data showing the effects on bacterial clearance is shown in Figure 10. Many of the earlier figures demonstrate that Th1 and Tfh are impaired in influenza infection but the effects of this on the resolution of infection are not shown. Does the CD28flox/flox OX40cre phenotype affect the natural history of influenza infection in these mice?

We have assessed influenza viral RNA levels in the lung seven days post infection, when the HKx31 is in the process of being cleared. We do not see a difference in viral clearance at this time point (included as a new Figure 7). As we did not observe a difference in viral load we assessed whether the flu specific CD8 response was intact in *Cd28*^*flox/flox*^
*Ox40*^*cre/+*^ mice. We found equivalent numbers of Dextramer-binding CD8^+^ T cells in *Cd28*^*flox/flox*^
*Ox40*^*cre/+*^ mice and control mice (Figure 7), suggesting that the CD8^+^ cells may be able to clear the virus even with a reduced population of CD4^+^ effector T cells. This is consistent with previous reports that CD4^+^ deficient animals are able to clear primary influenza infection.

*5) Studies in the Citrobacter model would benefit from a kinetic analysis of bacterial growth allowing assessment of the impact on early innate and later adaptive responses. It will be particularly important to look at ILC3 and Th17 cells responses as these are important components of host protective immunity to this enteric pathogen*.

In addition to the time course of *C. rodentium* shedding in the faeces presented in the original Figure 10A (now Figure 9) we have provided additional data showing the bacterial load in the colon, caecum and liver 7, 12 and 24 days following *C. rodentium* infection (Figure 9). These data show that *Cd28*^*flox/flox*^
*Ox40*^*cre/+*^ mice have comparable bacterial burdens to control mice in the early phase of infection but that subsequent clearance of *C. rodentium* is impaired.

The number of ILC3 and Th17 cells were determined seven days following infection. *Cd28*^*flox/flox*^
*Ox40*^*cre/+*^ mice had a comparable number of ILC3 cells to control mice, by contrast Th17 cells were reduced. These data have been included in Figure 9, and discussed in the text within the Discussion section.